# Structure of the endosomal CORVET tethering complex

Dmitry Shvarev [1,5], Caroline König [2,5], Nicole Susan [2,5], Lars Langemeyer [2,3], Stefan Walter [3], Angela Perz[2], Florian Fröhlich [3,4], Christian Ungermann [2,3] & Arne Moeller [1,3]

Cells depend on their endolysosomal system for nutrient uptake and down-regulation of plasma membrane proteins. These processes rely on endosomal maturation, which requires multiple membrane fusion steps. Early endosome fusion is promoted by the Rab5 GTPase and its effector, the hexameric CORVET tethering complex, which is homologous to the lysosomal HOPS. How these related complexes recognize their specific target membranes remains entirely elusive. Here, we solve the structure of CORVET by cryo-electron microscopy and revealed its minimal requirements for membrane tethering. As expected, the core of CORVET and HOPS resembles each other. However, the function-defining subunits show marked structural differences. Notably, we discover that unlike HOPS, CORVET depends not only on Rab5 but also on phosphatidylinositol-3-phosphate (PI3P) and membrane lipid packing defects for tethering, implying that an organelle-specific membrane code enables fusion. Our data suggest that both shape and membrane interactions of CORVET and HOPS are conserved in metazoans, thus providing a paradigm how tethering complexes function.

Eukaryotic cells maintain an elaborate endomembrane system of organelles. This interconnected network depends on the vesicular transport and relies on conserved types of machinery for vesicle generation at the donor organelle and fusion at the acceptor organelle[1–3]. For proper intracellular recognition, each organelle exhibits distinct membrane compositions and shapes. Identity markers include specific phosphoinositides (PIPs) in the lipid bilayer[4] and distinct peripheral membrane proteins such as Rab GTPases (Rabs)[3,5]. Together, these elements guide trafficking effector proteins to their specific membrane and enable the effective direction of vesicles to their acceptor organelles[4,6–8].

Rabs are an integral part of the membrane fusion cascade, as their depletion leads to impaired membrane trafficking[9,10]. As switch-like proteins, Rabs only interact with their effector proteins upon activation by guanine nucleotide exchange factors (GEFs) and binding to GTP while requiring a GTPase-activating protein (GAP) to hydrolyze GTP for their inactivation[3,5,11–13].

The most prominent effector proteins are tethering and fusion factors and complexes[14–18]. They tether the membranes and recruit Sec1/Munc18 proteins (SM proteins) to promote the zippering of SNAREs from each membrane into a four-helix bundle to trigger fusion[14,17–19].

The hexameric HOPS and CORVET are evolutionarily conserved tethering complexes within the endolysosomal system of eukaryotic cells[16,20–24]. Both share four subunits. Vps11 and Vps18 form the central core to which Vps33 and Vps16 are attached as an SM module to promote SNARE assembly[25,26]. The two unique subunits determine the respective Rab specificity of the tethering complexes[22–24,27–29]. In CORVET, Vps3 and Vps8 bind to Rab5 on early endosomes (EE)[22,23,27], while in HOPS, Vps41 and Vps39 interact with the Rab7-like Ypt7 in

[1]Department of Biology/Chemistry, Structural Biology Section, Osnabrück University, 49076 Osnabrück, Germany. [2]Department of Biology/Chemistry, Biochemistry Section, Osnabrück University, 49076 Osnabrück, Germany. [3]Center of Cellular Nanoanalytics Osnabrück (CellNanOs), Osnabrück University, 49076 Osnabrück, Germany. [4]Department of Biology/Chemistry, Bioanalytical Chemistry Section, Osnabrück University, 49076 Osnabrück, Germany. [5]These authors contributed equally: Dmitry Shvarev, Caroline König, Nicole Susan. ✉e-mail: cu@uos.de; arne.moeller@uos.de

yeast, and with Rab2 and lysosomal Arl8 in metazoan HOPS[16,18,20,21,30–33]. The specificity of the distinctive subunits is fundamental to the different roles of the two complexes within the endolysosomal system. CORVET detects Rab5, which decorates early endosomes and, therefore, functions in the fusion of endocytic vesicles and among EEs. HOPS promotes the fusion of late endosomes, autophagosomes, and vacuoles. Given their central position in the endolysosomal pathway, it is not surprising that CORVET[34] and HOPS have been linked to diseases and infections[16,35–39].

So far, most mechanistic insights into endolysosomal tethers have been obtained on the yeast HOPS complex. Proteoliposome assays and in vivo analyses revealed how it catalyzes fusion through tethering of Ypt7-decorated membranes and promoting the assembly of membrane-anchored SNAREs[40–44]. How CORVET works remains however unclear, and any discussions on similarity between HOPS and CORVET are speculative in the absence of structural insights.

Here, we present the structure of CORVET that resolves this puzzle. The overall structure of CORVET is conserved and highly similar to HOPS[45], while the Rab/membrane-interacting subunits, which control the respective functions of the two complexes, are remarkably different. Unlike HOPS, which solely depends on Ypt7 to tether membranes, CORVET requires PI3P and loose membrane packing in addition to its Rab5 GTPase for tethering. In combination, our study provides structural and functional evidence of how a multisubunit tethering complex perceives the combinatorial code of the membrane environment. As CORVET and HOPS share a similar shape

and composition, and their subunits are highly homologous throughout eukaryotes, we predict that analogous tethering complexes from other organisms, including humans, will be similar if not identical to the yeast complexes.

## Results

### Cryo-EM structure of the CORVET complex and its specific subunits

CORVET was produced in yeast cells and isolated via a FLAG tag on Vps8 using affinity chromatography followed by size exclusion chromatography (SEC) with high purity (Fig. 1A, B). The molecular weight of the purified complex measured by mass photometry (656 kDa) agreed with the predicted weight of 658 kDa (Fig. 1C).

Negative stain electron microscopy (EM) revealed evenly distributed single particles of CORVET with no apparent aggregates in the sample (Supplementary Fig. 1A), allowing further single-particle cryo-EM analysis (Supplementary Fig. 1B, 2). From 33882 micrographs, 219391 selected particles provided a consensus map with 4.6 Å resolution. Local refinements further improved the resolution to 3.8 Å (SNARE-binding module, Supplementary Fig. 3, Supplementary Table 3). A composite map of all local refinements (Fig. 1D, Supplementary Fig. 2) enabled model building of the CORVET structure (Figs. 1E, 2, Supplementary Fig. 4, Supplementary Movie 1, Supplementary Table 3).

CORVET and HOPS share strong structural homology. Their respective cores, consisting of identical subunits (Vps11 and Vps18), are virtually indistinguishable from each other (Fig. 2E). A similar

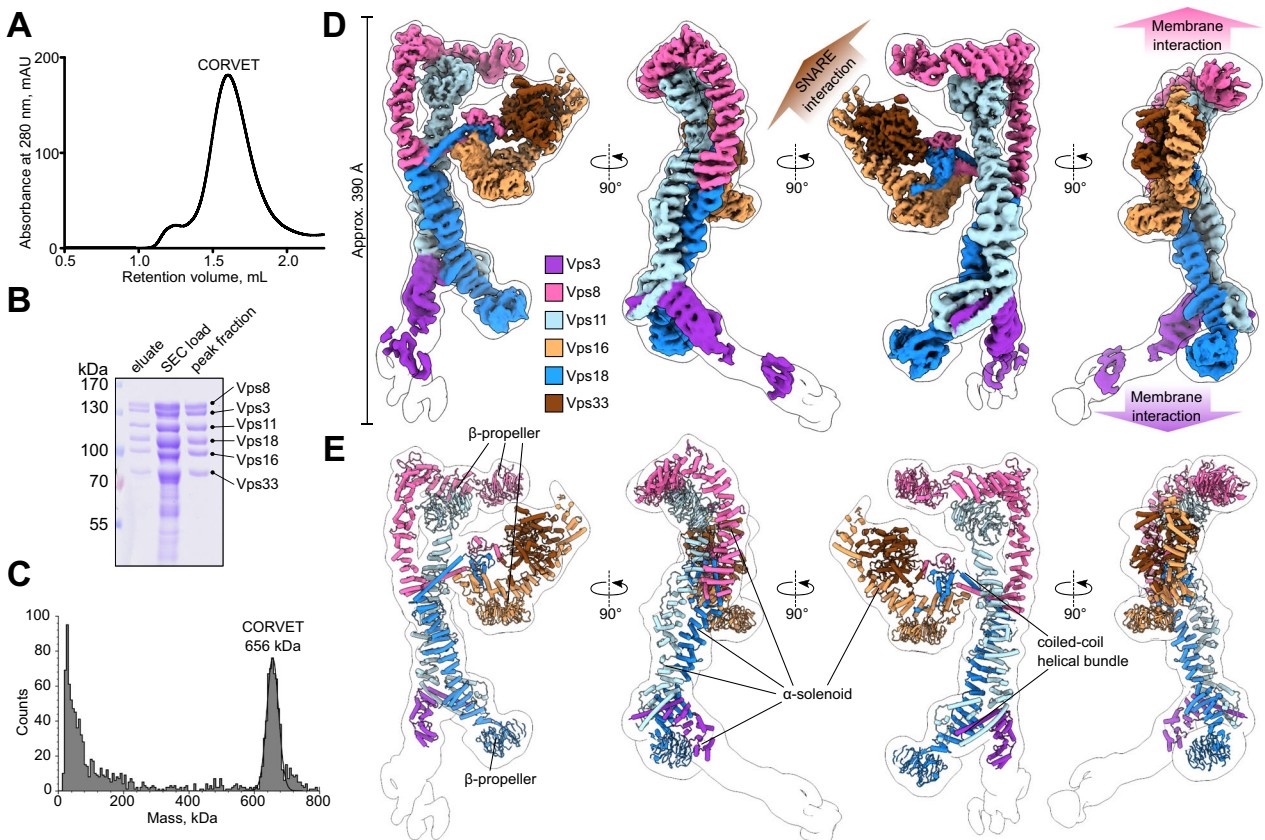

**Fig. 1 | Purification and cryo-EM structure of the yeast CORVET complex. A** Size exclusion chromatography (SEC) of the affinity-purified CORVET. The representative result of three independent experiments is shown. **B** SDS-PAGE analysis of purified CORVET. Protein samples from affinity purification (eluate) and before and after SEC are shown. *n* = 3 independent experiments were performed. **C** Mass photometry analysis of the peak fractions from SEC of CORVET. **D** Overall

architecture of the CORVET complex. Composite cryo-EM map generated from local refinement maps (see Figs. S2, S3) is colored by subunits assigned (Vps3, violet; Vps8, pink; Vps11, light blue; Vps16, sand; Vps18 blue; Vps33, brown). The consensus maps used for local refinements are low-pass-filtered and shown as a transparent envelope with a black outline. **E** Molecular model of CORVET fitted into the low-pass-filtered consensus map.

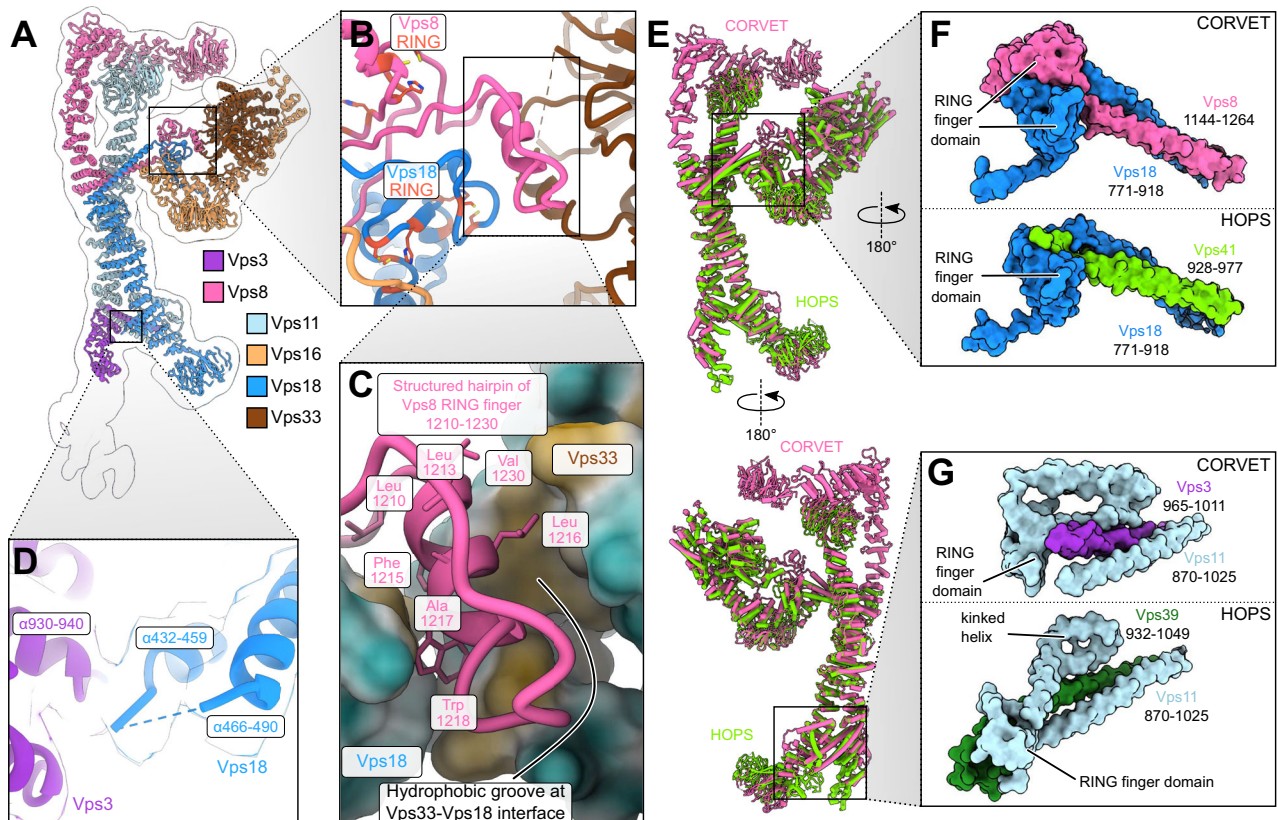

**Fig. 2 | Interactions of CORVET functional subunits with the core of the complex. A** Model of the CORVET complex viewed from the side in ribbon representation. The low-pass-filtered consensus map is shown by a black outline. The subunits are colored as in Fig. 1. **B** Zoomed-in view of the Vps33-Vps8-Vps16 interface with Vps8 and Vps18 RING finger domains indicated. **C** Zoomed-in view of the Vps8 structured hairpin sandwiched between Vps33 and Vps18. **D** Interactions between the α-solenoids of Vps18 and Vps3. Associated semi-transparent cryo-EM densities are zoned around the molecular models and colored accordingly. **E** Structural alignment of the CORVET molecular model (pink, this study) with the HOPS model (light green, PDB:7ZU0) viewed from two sides. **F** Close-up views of C-terminal two-helix bundle and RING finger domains of CORVET Vps8 and Vps18 (top) and HOPS Vps41 and Vps18 (bottom). **G** Close-up views of C-terminal two-helix bundle and RING finger domains of CORVET Vps3 and Vps11 (top) and HOPS Vps39 and Vps11 (bottom).

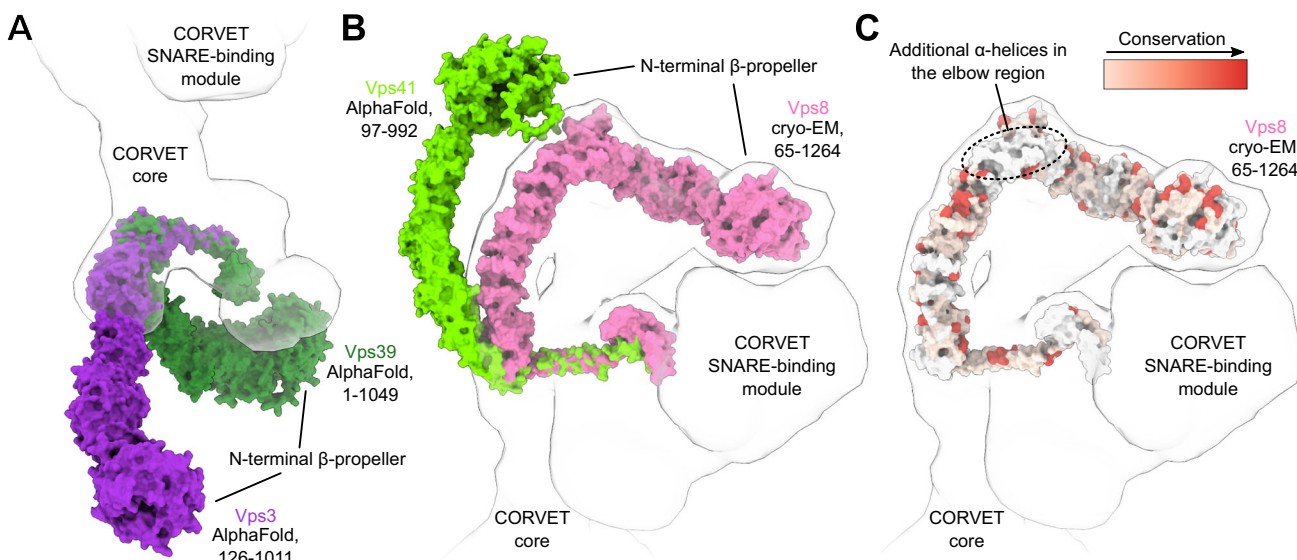

**Fig. 3 | Rab-binding subunits of CORVET compared to HOPS. A** Structural alignment of CORVET Vps3 (violet, AlphaFold prediction) with HOPS Vps39 (dark green, AlphaFold prediction). **B** Structural alignment of CORVET Vps8 (pink, cryo-EM structure from this study) with HOPS Vps41 (light green, AlphaFold prediction). **C** Vps8 subunit colored by conservation according to the sequence alignment with Vps41. In (**A**–**C**), the models of proteins are fitted into the semi-transparent volume (black outline) generated from the CORVET structure.

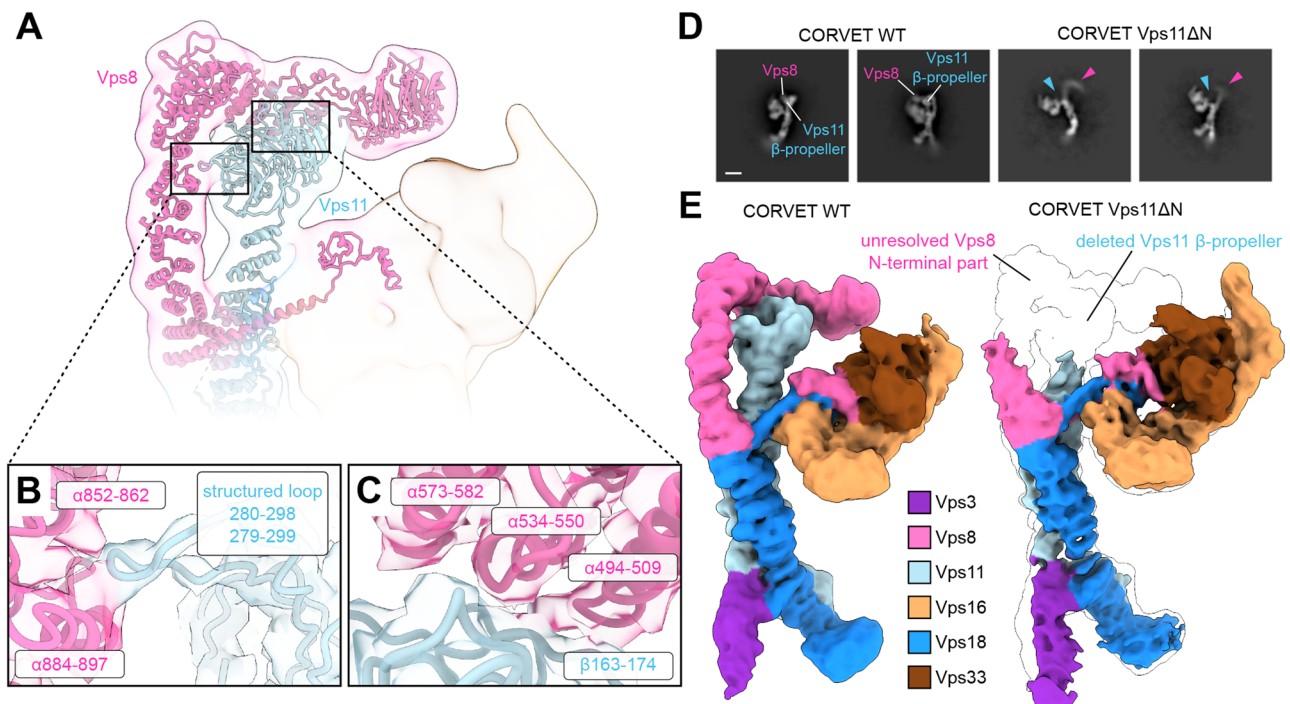

**Fig. 4 | CORVET-specific Vps8-Vps11 N-terminal interface. A** Molecular models of CORVET Vps8 and Vps11 in ribbon representation fitted into the semi-transparent low-pass-filtered consensus cryo-EM map colored by subunits as in Fig. 1. **B**, **C** Zoomed-in views of the interface sites between Vps8 and Vps11 with interacting structural elements indicated. Associated semi-transparent cryo-EM densities are zoned around the molecular models and colored accordingly. **D** Representative cryo-EM 2D class averages of CORVET wild-type and Vps11ΔN mutant. Structural differences in the regions of Vps8 and Vps11 N-termini are indicated. **E** Comparison of CORVET wild-type and Vps11ΔN mutant cryo-EM densities colored by subunits as in Fig. 1. The mutant cryo-EM density is overlaid on the wild-type density (transparent, black outline) highlighting the structural differences.

scenario is found for the SNARE-binding module (Vps33-Vps16) which branches off sideways from the core.

Reflecting their diverse functions within the endolysosomal system, those parts of the complexes that are responsible for target recognition show marked differences (Fig. 3). Vps3 and Vps8 are specific to CORVET and located at the distal ends of the complex (Fig. 1D, E). Each interlocks with the core through long C-terminal α-helices, reminiscent to the HOPS-specific Vps39 and Vps41 subunits (Figs. 1E, 2E–G, 3).

The extended Vps3 subunit is located at one end of the structure, but substantial flexibility in this region (Supplementary Fig. 5, Supplementary Movies 2–4) limited the achievable resolution. As such, the N-terminal part of Vps3, carrying the β-propeller, is only visible at lower thresholds (Fig. 1E, black outline). Similarly, to the analogous subunit Vps39 in HOPS, the C-terminal α-helix of Vps3 is anchored to CORVET's core through an interaction with the RING finger domain and the long α-helix located at the C-terminus of Vps11 (Fig. 2E, G). In HOPS the RING finger domain of Vps11 tightly interacts with the C-terminal portion of Vps39 that follows the long α-helix. In contrast, the CORVET subunit Vps3 lacks any further C-terminal extensions after the long α-helix, thus significantly reducing the contact area in the Vps3-Vps11 interface. Furthermore, we observe a high flexibility of Vps3 and a more distant positioning from the complex core, when compared to HOPS (Figs. 1D, E, 3A). Interestingly, mutations in the Vps3-Vps11 interface can have profound impacts on the complex. As shown by mass spectrometry, CORVET variants carrying mutations in Vps11 (*ups11-1, ups11-3*)[46] destabilize the complex (Supplementary Fig. 6A–F). In case of the *ups11-3* allele the attachment of the Vps3 subunit to the core is abolished (Supplementary Fig. 6C, D), whereas the *ups11-1* allele disassembles the entire complex (Supplementary Fig. 6E, F).

Vps8, the second Rab-binding subunit of CORVET, is located at the opposite end of the complex (Fig. 1D, E). Vps8 exhibits a characteristic elbow configuration, which turns its N-terminal half towards the β-propeller of Vps11. In the equivalent Vps41 subunit of HOPS, no such coordination is observed (Fig. 3B). In fact, Vps41 is curved in the opposing direction and exhibits higher flexibility than Vps8[45]. Importantly, Vps8 carries 282 additional amino acids compared to Vps41. Sequence alignment revealed that the main portion of these residues are inserted as 8 additional α-helices into the elbow region of Vps8 (Fig. 3C).

The attachment of Vps8 to the core of the complex is similar to Vps3 and established through the long C-terminal α-helices and RING finger domains of the Vps8 and Vps18 subunits (Figs. 1E, 2A, E, F). This C-terminal helical bundle formed by Vps18 and Vps8 is followed by a pair of semi-symmetrically interacting RING finger domains (Fig. 2F, upper panel), reminiscent to the Vps11-Vps39 interface of HOPS (at the opposing end of the complex, compare Fig. 2 panels F and G). Mutations in this interface (*ups18-1*) disassemble the entire complex (Supplementary Fig. 6G, H), similar to the Vps3-Vps11 interface (Supplementary Fig. 6E, F). The C-terminus of Vps8 not only anchors the subunit to the core but also contributes to the attachment of the SNARE-binding module (Fig. 2A–C), analogous to Vps41 in HOPS. Distinctively, the binding interface between the core and the SNARE-binding module is larger in CORVET, which apparently increases stability. As in HOPS, Vps33 interacts with the RING finger domain of Vps18, and the N-terminal portion of Vps16 α-solenoid connects with the Vps8-Vps18 C-terminal long α-helical bundle (Fig. 2A–C, E). Moreover, a hairpin protruding from the Vps8 RING finger domain is sandwiched in a hydrophobic groove between Vps33 and Vps18 (Fig. 2B, C), clamping the SNARE-binding unit from both sides together with the C-terminus of Vps18 (Fig. 2A, F upper panel). A similar

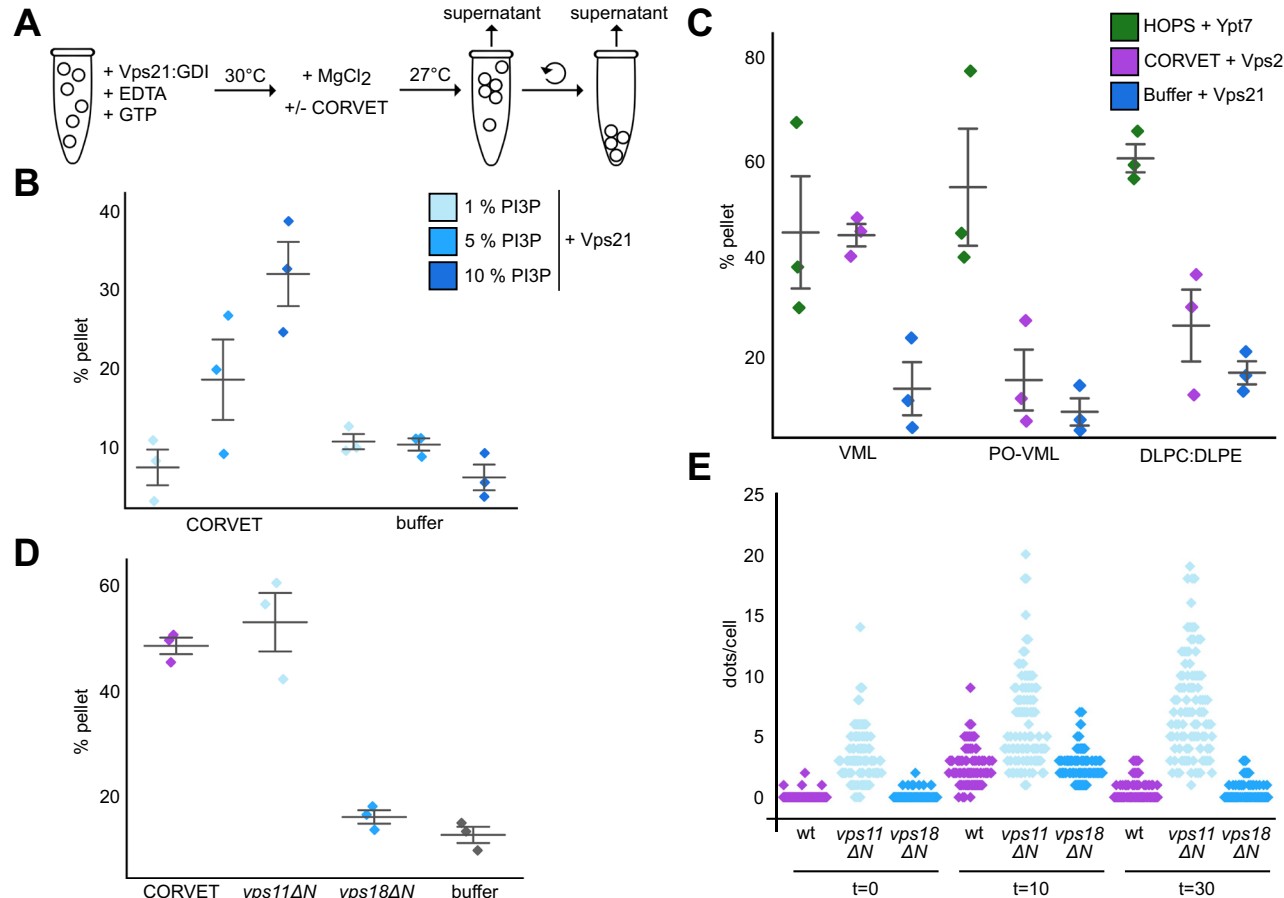

**Fig. 5 | Functional analysis of CORVET-mediated membrane tethering.**
**A** Scheme of the CORVET tethering assay (see "Methods" for details). **B** Tethering assay with phosphatidylinositol-3-phosphate (PI3P) titration. Prenylated Vps21 was incorporated in its GTP-loaded form into fluorescently labelled liposomes containing 1, 5, or 10 mol % PI3P. Liposomes were incubated with purified CORVET complex or a buffer control. Tethering activity was determined as described in materials and methods. Colored diamonds indicate single measurements. Bars indicate mean +/− 1 SE. Further tethering and statistics in Supplements. **C** Tethering assay with 1,2-dilinoleoyl-sn-glycero-3-phosphocholine (DLPC) and 1,2-dilinoleoyl-sn-glycero-3-phosphoethanolamine (DLPE) (DL-lipids) and palmitoyl/oleoyl (PO)-lipids. Liposomes containing 18:2/18:2 (DL-lipids) or 16:0/18:1 PO-lipids and L-α-phosphatidylinositol (SoyPI), PI3P, ergosterol and 1,2-dipalmitoyl-sn-glycero-3-(cytidine diphosphate) (DAG) or liposomes containing DLPC and DLPE and PI3P were fluorescently labelled. Prior to incubation with HOPS or CORVET liposomes were preloaded with prenylated and GTP-loaded Ypt7 or Vps21 out of the GDP

dissociation inhibitor (GDI) complex. Activity was measured according to materials and methods. Bars indicate mean +/− 1 SE. Colored diamonds indicate single measurements. **D** Tethering assay with CORVET carrying N-terminally truncated Vps11 and Vps18 subunits. Fluorescently labelled liposomes, decorated with prenylated and GTP-loaded Vps21, were incubated with wild-type or mutated CORVET variants. Tethering activity was measured as described in materials and methods. Bars indicate mean +/− 1 SE. Colored diamonds indicate single measurements. For all tethering assays (**B**–**D**), tree (*n* = 3) independent measurements were conducted for every condition tested. **E** Quantification of Mup1 uptake assay. Mup1 was GFP-tagged in wild-type, or cells expressing truncated *vps11ΔN* or *vps18ΔN*. Cells were grown in synthetic medium lacking methionine and imaged by fluorescence microscopy (*t* = 0), before shifting to media containing methionine for 10 min (*t* = 10) and 30 min (*t* = 30). The number of GFP-positive dots per cell is plotted for each strain. Colored diamonds indicate single cells (*n* = 100 cells were used for every experimental condition).

interface likely exists in human HOPS, where the analogous Vps41 subunit also has a RING finger domain, which is lacking in yeast Vps41[24,47].

### Structural and functional implications of Vps8 interactions with Vps11 β-propeller

The hallmark of CORVET is the Vps8 subunit with its elbow shaped configuration. In addition to the C-terminal interaction with Vps18, and distinctive from HOPS, the α-solenoid of Vps8 has two additional interfaces with Vps11 (Figs. 1D, E, 4A). The first interface is between two α-helices of Vps8 (residues 852-862 and 884-897) and a structured loop at the Vps11 β-propeller (residues 279-299) (Fig. 4B), which stabilizes the upright section of the α-solenoid before the elbow. The second interface after the elbow connects three α-helices of Vps8 (residues 494-509, 534-550, 573-582) and the outward side of Vps11 β-propeller (residues 163-174) (Fig. 4C).

As these interfaces are distinct to CORVET, we investigated them closely. A mutant complex lacking the Vps11 β-propeller (CORVET Vps11ΔN, residues 1-335) could be purified like the wild-type complex (Supplementary Fig. 7A, C). Cryo-EM analysis indicates that this mutant not only lacks the β-propeller of Vps11, but shows poor resolution of N-terminal part of Vps8 (Fig. 4D, E). We attribute this to increased flexibility in Vps8 due to the missing interface with Vps11.

We next explored if the corresponding mutation in CORVET affects its function. For HOPS, reconstituted tethering and fusion assays have been established[42,48,49], whereas comparable assays were missing so far for CORVET[22,42,50]. We thus set up a similar fluorescence-based tethering assay for CORVET (Fig. 5A) using liposomes loaded with Rab5(Vps21)-GTP[45,51,52], including a screen for membrane conditions. Initially, we used vacuole mimicking lipid mix (VML) established for vacuole fusion assays[53] for the generation of liposomes (see "Methods"). Due to the unsaturated C18:2 lipids in the phospholipids,

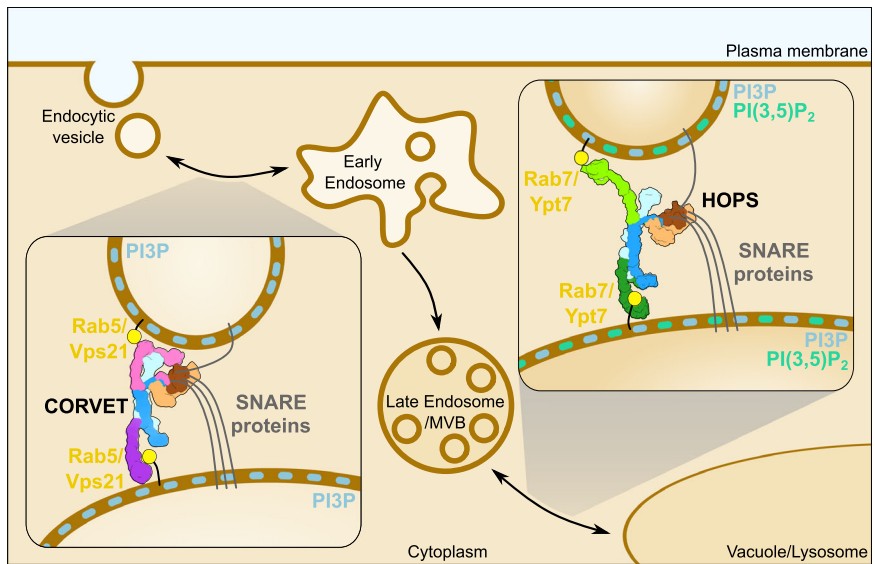

**Fig. 6 | Mechanistic model of membrane tethering by CORVET and HOPS within the endolysosomal pathway.** CORVET is responsible for early-endosomal fusion events and selectively relies on the Rab5 (Vps21) and phosphatidylinositol-3-phosphate (PI3P) presence in the membranes for function. In contrast, the lysosomal tether HOPS requires Rab7 (Ypt7) and, potentially, different lipid composition (e.g. phosphatidylinositol 3,5-bisphosphate (PI(3,5)P$_2$)) to promote membrane fusion. The cartoon, including the SNARE proteins, Rab GTPases, and membranes, is not drawn to scale.

the corresponding liposomes have a more fluid membrane with packaging defects. This analysis revealed that CORVET requires relatively high PI3P (10 mol%) concentrations for efficient tethering (Fig. 5B). To ask if membrane lipid packing defects contribute to efficient CORVET activity in tethering, we kept the VML composition for liposomes with the established PI3P concentration, but altered the acyl chain composition to palmitoyl (C16:0) oleoyl (C18:1) (PO) phospholipids (PO-VML), or used a more simple DL-phosphatidylcholine (PC) and DL-phosphatidylethanolamine (PE) mixture with the same PI3P concentration for liposomes (DLPC:DLPE). CORVET was only able to tether liposomes containing PI3P and the more fluid VML mixture, but none of the other liposomes (Fig. 5C). This contrasts to HOPS, where the GTP-loaded Rab7 GTPase Ypt7 is sufficient for tethering liposomes regardless of the composition (Fig. 5C). We thus conclude that CORVET requires multiple factors—PI3P, Vps21 and membrane packaging defects—to bind membranes.

We then compared wild-type CORVET to mutants lacking either the Vps11 or Vps18 β-propeller (Vps11ΔN and Vps18ΔN, respectively) in tethering assays. Surprisingly, the more flexible Vps11ΔN complex was still functional in tethering on membranes with high PI3P, whereas the Vps18ΔN was inactive (Fig. 5D). The Vps18ΔN was fully assembled as it was purified as a hexamer like the wild type (Supplementary Fig. 7B), suggesting that the Vps18 β-propeller itself is critical for efficient tethering.

To determine if mutations in the β-propeller of Vps11 or Vps18 affect CORVET function, we used corresponding cells for possible endolysosomal defects (Supplementary Fig. 8). We initially analyzed vacuole morphology as CORVET deletion mutants have enlarged vacuoles[54–56], but observed no difference to wild-type. However, cells expressing Vps11ΔN grew slower on plates containing endolysosomal stressors like ZnCl$_2$, indicating possible defects (Supplementary Fig. 8B). To analyze early endocytic trafficking, we followed the transport of the GFP-tagged methionine transporter Mup1 from the plasma membrane via endosomal dots to the vacuole (Fig. 5E, Supplementary Fig. 8C). Whereas Mup1 was efficiently transported to the vacuole in wild-type and Vps18ΔN cells, it accumulated in GFP-positive dots over long periods in Vps11ΔN cells, suggesting impaired endosomal transport toward the vacuole in this mutant in vivo.

## Interaction of CORVET with membranes

To analyze how CORVET interacts with membranes, we first used in silico prediction of interactions of CORVET's subunits Vps8 or Vps3 with the Rab5 GTPase Vps21. Our AlphaFold modeling[57] of the Vps3-Vps21 and Vps8-Vps21 interactions (Supplementary Fig. 9A–F) suggested binding sites on the peripheral areas of Vps8 and Vps3 α-solenoids, which would be accessible by membrane-bound Vps21 via its 10 nm long hypervariable domain (not shown in the prediction). The predicted interaction mode is similar to that in the HOPS complex[45], involving analogous conserved residues localized to the switch regions in the Rab GTPases (Supplementary Fig. 9A–G), which are known to mediate Rab-effector binding[12,58].

To visualize CORVET on membranes, we incubated the complex with PI3P-containing liposomes loaded with Vps21-GTP and analyzed these by negative-stain microscopy. We observed CORVET particles coating the vesicle surfaces, mostly in an upright position (Supplementary Fig. 9H). Some particles also appeared flat on the membranes, possibly via Rab binding of both membrane-binding subunits in CORVET. The liposomes incubated without CORVET were indeed free of any decoration (Supplementary Fig. 9I). We conclude that CORVET is like HOPS on membranes preferentially upright prior to tethering.

## Discussion

In this study, we solved the structure of the crucial metazoan endolysosomal tethering complex CORVET. We show that CORVET and HOPS have modular architecture sharing an identical core with the attached SNARE-binding module and distinct complex-specific membrane-binding subunits. Importantly, we uncovered that CORVET differs strongly from HOPS in that it requires (i) PI3P, (ii) membrane packaging defects, and (iii) Rab5 (Vps21) for its efficient tethering function. Our data suggest that tethering complexes thus read out their membrane environment at several levels to gain organelle-specificity (Fig. 6).

As in HOPS, subunit interactions in CORVET rely on the presence of C-terminal long α-helices and RING finger domains. As CORVET Vps3 misses a C-terminal RING finger domain, its association is weaker than Vps39 binding in HOPS, at least in solution[22,28]. This is confirmed by our analysis of the Vps11-specific mutant alleles, which trigger CORVET disassembly in vivo (Supplementary Fig. 6C–F). It has been

shown that RING finger domains of HOPS and CORVET subunits have ubiquitin ligase activity[59]. As RING finger domains are structural elements of HOPS and CORVET, they may function as quality control elements to monitor the assembly of the complex. This could also explain, why an intermediate complex between HOPS and CORVET exists, which maintains the core part and can swap functional subunits such as Vps41/Vps8 or Vps39/Vps3, while the free subunit may fulfill auxiliary functions[22,60]. When Vps8 alone is overproduced, cells accumulate MVBs proximal to the vacuole[27], suggesting that Vps8 can constitute a minimal tether. Interestingly, a minimal CORVET complex, lacking Vps11 as a central subunit, was proposed in *Drosophila*[61], which is, however, incompatible with the identified Vps11-Vps18 core of CORVET and HOPS. It is thus not yet clear if miniCORVET is indeed a minimal CORVET or may have additional subunits.

One striking difference between CORVET and HOPS is the rigid apposition of Vps8 to the Vps11 β-propeller. This tight interaction can explain, why overproduced Vps8 in *Drosophila* can outcompete Vps41 and thus prevent HOPS assembly as it will likely sequester the available Vps11 pool[62]. The Vps8 apposition to the core and the Vps8-Vps11 contacts appear to be important for full CORVET activity in early endocytic transport in vivo. Interestingly, a reported point mutation in the α-solenoid of Vps8 might cause a destabilizing of the Vps8-Vps11 interaction[29]. We demonstrate this by our structural (Fig. 4) and functional (Fig. 5) analyses of CORVET Vps11ΔN. Indeed, a deletion of the β-propeller domain of core subunits such as Vps11 and Vps18 will affect both CORVET and HOPS in vivo. In this regard, we were surprised that the CORVET Vps18ΔN mutant, in contrast to Vps11ΔN, was largely inactive in the tethering assay (Fig. 5D), whereas the comparable HOPS complex was functional[45]. This suggests that Vps18 either contributes to membrane recognition by Vps3 within CORVET or to the structural integrity of the complex. Curiously, in contrast to the in vitro results, introduction of the *vps18ΔN* mutation did not have an effect in vivo, while the *vps11ΔN* mutation caused significant endosomal defects (Fig. 5E, Supplementary Fig. 8). This difference may be due to the fact that in vivo, the *vps11ΔN* mutation impacts not only CORVET but also HOPS, while the *vps18ΔN* mutant may be complemented by additional cellular factors not included in our in vitro assays. Further precise analyses, such as introducing point mutations into specific subunits of CORVET and HOPS, and establishing an in vitro fusion assay for CORVET, will help to better explain the discrepancies between the in vitro and in vivo situations, as well as the specific functional differences between HOPS and CORVET. In either case, we can only speculate, how membranes are decoded by CORVET and how the β-propeller domains of Vps11 and Vps18 contribute. One idea is that the β-propeller of the core subunits Vps11 and Vps18 bind Rab5 GTPases together with the Rab5-GTP specific subunits Vps3 and Vps8, thus optimally positioning them for direct PI3P and membrane binding.

We show that CORVET depends on multiple membrane cues, whereas HOPS can tether any membrane carrying the Rab GTPase Ypt7. For membrane binding, CORVET may thus require specific lipids and membrane packaging defects, either through high curvature or due to the membrane composition[6,63], in addition to Rab5, which is needed for CORVET localization in vivo[55]. In line with this, predicted Rab binding sites of CORVET and HOPS are largely conserved and located at the distal exposed ends of the complexes. However, Rab5 also binds and stimulates the Vps34 PI-3-kinase complex, which establishes a suitable membrane environment required for CORVET recognition[64,65]. Consequently, loss of Rab5 (or its homolog in yeast) will possibly affect both the CORVET binding and the specific membrane environment.

Currently, we have no clear understanding, how tethering complexes function on membranes to promote tethering and fusion. Our analyses of HOPS suggest that the complex is positioned upright on membranes when bound to the Rab7-like Ypt7 protein[66]. Here, we used

liposomes and find CORVET also mostly upright on membranes (Supplementary Fig. 9H). This suggests that direct interactions with the lipid bilayer strongly contribute to the positioning of CORVET, making it ready for tethering and subsequent promotion of SNARE-mediated fusion, while Rab GTPases may further stabilize the complex.

Our analysis of CORVET reveals that its core architecture largely resembles HOPS, yet the orientation and structure of the Rab-specific subunits and their decoding of membranes strongly differs between both complexes. Nevertheless, though interchangeable assemblies of CORVET and HOPS are possible, the four shared central subunits, Vps11, 18, 33 and 16, are found in all species[67]. We are thus convinced that the overall arrangement and function of HOPS and CORVET as a modular membrane fusion machinery is evolutionary conserved. Future studies will take advantage of our structural insights to determine, how these complexes promote SNARE-mediated membrane fusion.

## Methods
### Yeast strains
Yeast strains and oligonucleotides used in this study are listed in Supplementary Table 1 and Supplementary Table 2 respectively. In general, CORVET subunits were expressed under the control of the GAL1 promoter according to the standard protocol[68]. For subunit truncation (Vps11 or Vps18) of the N-terminal part, the GAL1 promotor was inserted into the genome at the respective position. The 3x-FLAG Tag was attached to the CORVET subunit Vps8, except for the Vps18ΔN mutant.

### Protein expression and purification from *Escherichia coli*
Rab GTPases for tethering assays were expressed in *Escherichia coli* BL21 (DE3) Rosetta cells. Cultures were grown in Luria broth (LB) medium complemented with 35 μg/ml kanamycin or 100 μg/ml ampicillin and 30 μg/ml chloramphenicol. Cultures were induced with 0.5 mM Isopropyl-β-D-thiogalactoside (IPTG) and incubated overnight at 16 °C. Cells were harvested by centrifugation (4800 × $g$, 10 min, 4 °C) and resuspended in buffer (150 mM NaCl, 50 mM HEPES/NaOH, pH 7.4, 10% glycerol, 1 mM PMSF, and 0.5-fold protease inhibitor mixture [PIC]) prior to lysis in a Microfluidizer (Microfluidics Inc). Crude lysates were centrifugated at 25,000 × $g$, 30 min, 4 °C. Supernatants were incubated with glutathione Sepharose (GSH) fast flow beads (GE Healthcare) for GST-tagged proteins or nickel–nitriloacetic acid (Ni-NTA) agarose (Qiagen) for His-tagged proteins (2 h, 4 °C). Proteins were eluted with buffer (150 mM NaCl, 50 mM HEPES/NaOH, pH 7.4, 10% glycerol) containing either 25 mM glutathione or 300 mM imidazole. Buffer was exchanged via a PD10 column (GE Healthcare). For tag cleavage, TEV or PreScission protease was added after washing and incubated overnight. Proteins were stored at −80 °C.

### Purification of 3xFLAG-tagged CORVET complex variants
CORVET tethering complex variants were purified according to the standard FLAG-purification protocol as previously described[45] with minor changes. Two liters medium (yeast peptone (YP), containing 2% galactose (v/v)) were inoculated with 6 ml of an overnight preculture. Cultures were grown for 24 h and harvested by centrifugation (4800 × $g$, 10 min, 4 °C). Cell pellets were washed with cold CORVET purification buffer (CPB, 300 mM NaCl, 20 mM HEPES/NaOH, pH 7.4, 1.5 mM MgCl_2, and 10% (v/v) glycerol). Pellets were resuspended in CORVET lysis buffer (CLB, CPB supplemented with 1 mM phenylmethylsulfonylfluoride (PMSF), 1× FY protease inhibitor mix (Serva) and 1 mM dithiothreitol (AppliChem GmbH)). Cell suspension was dropwise frozen in liquid nitrogen before lysis in a freezer mill cooled with liquid nitrogen (SPEX SamplePrep LLC). For purification, the powder was thawed on ice and resuspended in CLB, followed by two steps of centrifugation at 5000 and 15,000 × $g$ at 4 °C for 10 and 20 min. The supernatant was combined with anti-FLAG M2 affinity gel

(Sigma–Aldrich) and placed on a nutator for 45 min at 4 °C. Beads were centrifugated (500 × g, 1 min, 4 °C) and transferred to a 2.5 ml MoBiCol column (MoBiTec). Samples were washed with 25 ml CPB before FLAG-peptide was added, followed by incubation on a turning wheel for 40 min at 4 °C. The eluate was concentrated in a Vivaspin 100 kDa MWCO concentrator (Sartorius), which was incubated with CPB containing 1% TX-100. Concentrated sample was applied to a Superose 6 Increase 15/150 column (Cytiva) for size exclusion chromatography (SEC). Fractions were eluted in 50 µl using ÄKTA go purification system (Cytiva). Peak fraction was used for further analysis.

## Mass photometry analysis

Mass photometry experiments were performed using a Refeyn TwoMP (Refeyn Ltd). Data were acquired using AcquireMP software and analyzed using DiscoverMP v2023 (both Refeyn Ltd). High Precision Cover Glasses (Marienfeld) were used for sample analysis. Perforated silicone gaskets were placed on the coverslips to form wells for every sample to be measured. Samples were evaluated at a final concentration of 10 nM in a total volume of 20 µl in the buffer used for SEC. Calibration was performed using β-amylase (Carl Roth).

## ALFA pulldowns for mass spectrometry

Vps8-ALFA pulldowns were performed in the same way as previously described[45]. One liter YP medium containing 2% glucose (v/v) was inoculated with an overnight preculture. Cells were grown to $OD_{600}$ 1 at 26 °C, followed by 1 h heat shock at 38 °C. Cells were harvested by centrifugation (4800 × g for 10 min at 4 °C). Pellets were washed with cold Pulldown buffer (PB, 150 mM KAc, 20 mM HEPES/NaOH, pH 7.4, 5% (v/v) glycerol and 25 mM CHAPS). Cells were resuspended in a 1:1 ratio (w/v) in PB supplemented with Complete Protease Inhibitor Cocktail (Roche) and afterward dropwise frozen in liquid nitrogen before lysis in 6875D LARGE FREEZER/MILL (SPEX SamplePrep LLC). Powder was thawed on ice and resuspended in PB, followed by two centrifugation steps at 5000 and 15,000 × g at 4 °C for 10 and 20 min. Supernatant was added to 12.5 µl prewashed ALFA Selector ST beads (2500 × g, 2 min, 4 °C) (NanoTag Biotechnologies) and incubated for 15 min at 4 °C while rotating on a turning wheel. Afterwards, beads were washed two times in PB and four times in PB without CHAPS. Samples were digested and prepared according to the PreOmics ST 96x Kit (iST Kit, preomics), using LysC as protease. Dried samples were resuspended in 10 µl LC-Load. 1 µl of the final sample was analysed in reversed-phase chromatography, performed on a Thermo Ultimate 3000 RSLCnano system, connected to a Q ExactivePlus mass spectrometer (Thermo Fisher Scientific) through a nano-electrospray ion source. 50 cm PepMap C18 easy spray columns (Thermo Fisher Scientific) with an inner diameter of 75 µm were used and kept at 40 °C. Peptides were eluted with a linear gradient of acetonitrile from 10 to 35% in 0.1% formic acid for 118 min at a constant flow rate of 300 nl/min, which was followed by electrospraying into the mass spectrometer. The mass spectra were acquired on the Q-Exactive Plus. The maximum injection time was set to 50 ms, the target value to 3,000,000 at a resolution of 70,000 at m/z = 200. From this, the ten most intense multiply charged ions (z = 2) from the survey scan were selected with an isolation width of 1.6 Th as well as fragments with higher energy collision dissociation with normalized collision energies of 27[69]. Target values for MS/MS were set at 100,000, with maximum injection time of 80 ms at a resolution of 17,500 at m/z = 200. The dynamic exclusion of the overall sequenced peptides was set at 20 s to avoid repetitive sequencing. The resulting MS spectra were analyzed using MaxQuant (v2.0.3.0, www.maxquant.org/)[70,71]. Data were analyzed using Persus (v2.0.11, www.maxquant.org/persus)[72]. Significances in the volcano plot of the Perseus software package corresponds to a given FDR which was determined by a permutation-based method[73].

## Negative stain analysis

Samples of wild-type CORVET, mutant CORVET variants, and CORVET incubated with liposomes were examined by negative-stain electron microscopy. 3 µl of sample at a protein concentration of approximately 0.05 mg/ml was applied onto freshly glow-discharged carbon-coated copper grids with plastic support, blotted from the side and stained using 2% (w/v) uranyl formate solution as previously described[74]. Negative-stain micrographs were collected manually on a JEM-2100Plus transmission electron microscope (JEOL) operated at 200 kV and equipped with a XAROSA CMOS 20-megapixel camera (EMSIS) at a nominal magnification of 30,000 (3.12 Å per pixel). The data was analyzed using ImageJ v1.52k[75] and cryoSPARC v3 and v4[76].

## Cryo-EM sample preparation and data acquisition

For cryo-EM, 3 µL of freshly purified CORVET samples at a protein concentration of approximately 0.8 mg/ml were applied to glow-discharged C-Flat grids (R1.2/1.3 3Cu-50) (EMS) and immediately plunge frozen in liquid ethane using a Vitrobot Mark IV (Thermo Fisher Scientific) with the environmental chamber set at 100% humidity and 4 °C.

Micrographs were collected automatically with EPU v3.7.1. (Thermo Fisher Scientific), using a Glacios cryo-transmission electron microscope (Thermo Fisher Scientific) operated at 200 kV and equipped with a Selectris energy filter and a Falcon 4 detector (both Thermo Fisher Scientific). Data were recorded in Electron Event Representation (EER) mode at a nominal magnification of 130,000 (0.924 Å per pixel) in the defocus range of −0.8 to −1.8 µm with an exposure time of 7.50 s resulting in a total electron dose of approximately 50 e⁻ Å⁻².

## Cryo-EM image processing

All cryo-EM data were preprocessed in cryoSPARC Live, and further processing was performed in cryoSPARC v3 and v4[76] (Supplementary Fig. 2). For all collected movies, motion correction (EER upsampling factor 1, EER number of fractions 40) and contrast transfer function (CTF) estimation were performed using cryoSPARC Live implementations. After micrograph curation, 33,882 micrographs were included for further data analysis.

For the determination of the wild-type structure, all collected datasets were preprocessed in a similar manner and combined at the final steps of image processing (Supplementary Fig. 2). Briefly, particles were selected by the template picker implemented in CryoSPARC Live, using well-defined 2D classes obtained from preliminary CORVET datasets as templates, and additional particle picking was performed using the Topaz wrapper[77]. For particle picking, micrographs from sessions 1–5 were combined, while the micrographs from sessions 6 and 7 were used separately. After particle duplicate removal and particle extraction in a box size of 882 pixels (Fourier-cropped to 128 pixels, resulting in a pixel size of 6.37 Å per pixel), two rounds of 2D classification were performed for all particle stacks obtained from each of the picking jobs to eliminate bad picks (Supplementary Fig. 2). After 2D classifications, duplicate particles were removed again and a round of ab-initio reconstruction with 5 classes followed by heterogeneous refinement was performed for each stack of particles resulted from sessions 1–5, session 6, and session 7. Each of these three heterogeneous refinement jobs produced one best class revealing all of the subunits of CORVET in the map. Such a class resulted from session 7 was further refined individually using NU refinement[78] and particles from it (158,046 particles) were afterwards combined with the particles from the best classes of the heterogeneous refinement jobs from sessions 1-5 (93661 particles) and 6 (216338 particles). A round of heterogeneous refinement with three classes was performed using these combined particles in a box size of 882 pixels (Fourier-cropped to 224 pixels, pixel size 3.64 Å per pixel), which resulted in two good classes reaching resolution of 7.4 Å (Gold Standard Fourier Shell

Correlation (GSFSC) value of 0.143). These two classes (174,263 and 198,464 particles) were combined and further refined using NU refinement in a box size of 882 pixels (Fourier-cropped to 400 pixels, pixel size 2.04 Å per pixel). The obtained reconstruction resolved to 4.6 Å (GSFSC = 0.143) was further classified using a round of heterogeneous refinement with 5 classes. The best class that reached resolution of 6.7 Å (GSFSC = 0.143) was used for a NU refinement providing a consensus map (219391 particles) at 4.6 Å resolution (GSFSC = 0.143). This consensus map was further used for local refinements of distinct areas of the structure. For local refinements, five masks were generated using UCSF ChimeraX 1.7.1[79] and particles were again re-extracted for some of the parts of the structure in a box size of 882 pixels (Fourier-cropped to 512 pixels, pixel size 1.59 Å per pixel) (Supplementary Fig. 2). A composite map was generated from the local refinement maps using the "volume maximum" command in UCSF ChimeraX.

All maps were subjected to unsupervised B-factor sharpening within cryoSPARC. No symmetry was applied during processing. The quality of the consensus and local refinement maps is shown in Figs. S3 and S4. All GSFSC curves and angular distribution plots were generated within cryoSPARC (Supplementary Fig. 3). The local resolutions of the consensus and local refinement maps (Supplementary Fig. 3) were estimated in cryoSPARC and analyzed in UCSF ChimeraX. Dataset statistics are provided in Supplementary Table 3. The cryo-EM analysis of the CORVET Vps11ΔN mutant was performed similarly to the wild type (Supplementary Fig. 7D).

### Model building and refinement
The structures of Vps11, Vps16, Vps18, and Vps33 from the structure of the HOPS complex (PDB: 7ZU0) and the AlphaFold structure predictions of Vps8 and Vps3 (Uniprot: P39702/P23643 respectively) were manually fitted into the consensus and local-refinement maps using the "Fit in Map" tool in UCSF ChimeraX and used as a starting model. Most of the structure was modelled as poly-alanine sequences using the PDB Tools job in PHENIX 1.20[80], except for the regions of the structure with the highest resolution obtained (SNARE-binding module, Vps8 α-solenoid, N-terminal half of Vps11). In addition, several protein fragments without well resolved corresponding EM densities were removed from the model (including the N-terminal half of Vps3). The models of the CORVET subunits were then manually adjusted and refined in Coot[81] and combined into a single model of the full complex. Subsequently, iterative rounds of real space refinement[82] of the model against the composite map of CORVET in PHENIX were performed, followed by manual adjustments in Coot 0.9.6. Model validation was done using MolProbity[83] in PHENIX. Models and maps were visualized, and figures were prepared in UCSF ChimeraX and Inkscape v1.3. Model refinement and validation statistics are provided in Supplementary Table 3.

### Liposome preparation
For the preparation of liposomes, the following lipids were used. 1,2-dilinoleoyl-sn-glycero-3-phosphocholine (18:2 PC), 1-palmitoyl-2-oleoyl-glycero-3-phosphocholine (POPC) 1,2-dilinoleoyl-sn-glycero-3-phosphoethanolamine (18:2 PE), 1-palmitoyl-2-oleoyl-sn-glycero-3-phosphoethanolamine (POPE), L-α-phosphatidylinositol (SoyPI), 1,2-dilinoleoyl-sn-glycero-3-phospho-L-serine (18:2 PS), 1-palmitoyl-2-oleoyl-sn-glycero-3-phospho-L-serine (POPS), 1,2-dilinoleoyl-sn-glycero-3-phosphate (18:2 PA), 1-palmitoyl-2-oleoyl-sn-glycero-3-phosphate (POPA), 1,2-dipalmitoyl-sn-glycero-3-(cytidine diphosphate) (DAG) were purchased from Avanti Polar Lipids (Alabama, USA). Phosphatidylinositol 3-phosphate (PI3P) was purchased from Echelon Biosciences Inc. ATTO488-1,2-dipalmitoyl-sn-glycero-3-phosphoethanolamine (ATTO488) was obtained from ATTO-TEC GmbH. Ergosterol was purchased from Sigma Aldrich.

For tethering assays and negative stain, lipid films, containing 37.6 or 42.6 mol% 18:2 PC/ POPC, 18 mol% 18:2 PE/ POPE, 18 mol% SoyPI, 10 or 5 mol% PI(3)P, 4.4 mol% 18:2 PS/ POPS, 2 mol% 18:2 PA/ POPA, 8 mol% Ergosterol, 1 mol% DAG and 1 mol% ATTO488, were evaporated by a SpeedVac (CHRIST). Lipid films prepared for the negative stain EM analysis did not contain dye and the 18:2 PC content was raised accordingly. Lipid films were resuspended in buffer (25 mM HEPES/NaOH, pH 7.4, 135 mM NaCl). Unilamellar vesicles were obtained through seven freeze and thaw cycles in liquid nitrogen. Liposomes were extruded through polycarbonate filters to 100 nm for tethering assays and 30 nm for negative stain EM analysis (400, 200, 100, 50 and 30 nm pore size) using a hand extruder (Avanti Polar Lipids, Inc.).

### Tethering assay
CORVET-mediated tethering assays were done as before[42]. ATTO488 labelled liposomes were prepared and loaded with prenylated Vps21 or Ypt7[52]. 100 nmole liposomes were incubated with 100 pmole pVps21:GDI/Ypt7:GDI in the presence of 1 mM GTP and 20 mM EDTA for 30 min at 30 °C, before the addition of 25 mM $MgCl_2$. 100 nM CORVET/HOPS or buffer (HEPES/NaOH, pH 7.4, 300 mM NaCl, 1.5 mM $MgCl_2$) were incubated with 0.17 mM Rab-loaded liposomes for 10 min at 27 °C. Liposomes were sedimented by centrifugation at $2500 \times g$ for 5 min. The tethered liposomes in the pellet fraction were determined by comparing the ATTO488 fluorescent signal of the supernatant before and after centrifugation using a SpectraMax M3 fluorescence plate reader (Molecular Devices). Statistical analyses were performed with Origin 2023 v10.0.

### Imaging of yeast cells using fluorescence microscopy
Cells were grown to the exponential phase in synthetic complete medium supplemented with all amino acids (SDC), or lacking methionine (SDS-met) [0.675% (w/v) yeast nitrogen base without amino acids, 2.0% (w/v) glucose, 0.075% (w/v) CSM (MPBiomedicals)]. Cells were imaged in SDC or SDC-met using the Zeiss Axioscope 5, with a PLAN-Apochromat 100x/1.40 Oil DIC M27 objective and an axiocam 702 mono (1.0x) camera.

All images were processed and quantified using ImageJ. Statistical analyses were performed with Origin 2023 v10.0 software. Vacuoles were stained using 30 μM FM4-64 (Molecular Probes Inc., Eugene, OR) for 30 min. Cells were washed twice with medium, prior to 1 h incubation at 30 °C[84].

### Reporting summary
Further information on research design is available in the Nature Portfolio Reporting Summary linked to this article.

### Data availability
The data that support this study are available from the corresponding authors upon request. The cryo-EM maps have been deposited in the Electron Microscopy Data Bank (EMDB) under accession codes EMD-18701 (CORVET composite map); EMD-18702 (CORVET Vps8-Vps11 local refinement map); EMD-18703 (CORVET Vps8 β-propeller local refinement map); EMD-18704 (CORVET SNARE binding module local refinement map); EMD-18705 (CORVET core local refinement map); EMD-18706 (CORVET Vps18 β-propeller local refinement map); EMD-18707 (CORVET consensus map); EMD-18708 (CORVET Vps11ΔN mutant map). The atomic coordinates have been deposited in the Protein Data Bank (PDB) under accession code 8QX8 (CORVET tethering complex). The previously published structure of HOPS tethering complex with the accession code 7ZU0 has been used in this study. The source data underlying Figs. 1a–c, 5b–e and Supplementary Figs. 6b–h, 7a–c, 8 are provided as a Source Data file. Source data are provided with this paper.

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

## Acknowledgements

The authors thank Jannis Schoppe for help with the initial purifications of the CORVET sample, Dovile Januliene for help with cryo-EM experiments, Kilian Schnelle for help with computing and data processing, and all members of the Ungermann and Moeller lab for feedback. This work was supported by a grant of the DFG (UN111/5-6; MO2752/3-6), the SFB 944 and SFB 1557 (C.U.; F.F., A.M.), the DFG INST190/196-1 FUGG (A.M.) and the BMBF/DLR 01ED2010 (A.M.).

## Author contributions

D.S., C.K. and N.S. conceived and designed all experiments together with A.M. and C.U. D.S. conducted all cryo-EM analyses. C.K. and N.S. did all biochemical and cell biology. experiments. F.F. performed mass

spectrometry analysis with support of S.W. A.P. gave technical support on biochemical and cell biology experiments. L.L. gave conceptional and technical advice on biochemical and cell biology experiments. The manuscript was written by D.S., C.U. and A.M. with contributions from all authors.

## Funding

## Competing interests
The authors declare no competing interests.
