## [Peer Review File · Nature Communications]

Structure of the endosomal CORVET tethering complexReviewer #1 (Remarks to the Author):

The authors present the first cryoEM structure of the endosomal tethering complex CORVET. The resolution of the reconstruction is good enough that most of the complex can be built, excepting a highly flexible portion of Vps3. The authors are then able to compare and contrast the structure of CORVET to that of the related lysosomal tethering complex HOPS. They are able to explain how these structural differences correspond to the known functional differences between the complexes.

The cryoEM model is consistent with previously published data on Vps11 and Vps18 ts-alleles, and the authors make new truncation mutations based on the structure in order to probe the consequences of these mutations. They find that a mutant lacking the Vps11 beta-propeller increases the flexibility of Vps8 by determining an additional cryoEM structure.

The authors establish an in vitro vesicle tethering assay for CORVET, similar to the one established for the related complex HOPS. They use the assay to determine that CORVET-dependent liposome tethering requires Rab5/Vps21, PI3P, and lipid-packing defects.

They then determined that the Vps18 beta-propeller was required for CORVET function in the tethering assay, but the Vps11 beta-propeller was not. In contrast, they find that the Vps11 beta-propeller is required for CORVET function in cells but the Vps18 beta-propeller was not.

Finally, the authors present AlphaFold predictions of Rab5/Vps21 interactions with CORVET along with lower-resolution (negative stain) structural information of CORVET bound to liposomes via interactions with Rab5/Vps21, providing evidence for an "upright" positioning of CORVET on endosomal membranes.

Overall I found this to be an interesting and well-performed study. Given the fundamental importance of CORVET in cell biology, I think this should appeal to a broad readership. I have only minor suggestions for improvement.

1. The authors do not directly address the discrepancies between the results of the in vitro tethering assay and the in vivo functional tests regarding the Vps11 and Vps18 beta-propellers. One interpretation is that the in vitro tethering assay does not faithfully represent the situation in vivo. And while no in vitro assay can be expected to fully recapitulate all aspects of in vivo function, the authors are encouraged to mention in the discussion possible reasons for this discrepancy and potential avenues for improving the assay in the future.

2. Lines 30 and 239 – "membrane packaging defects" should be "membrane packing defects" or maybe "lipid packing defects"

3. In Extended Figure S6C, E, and G, labels should be added for each subunit shown. I was initially confused because I thought the purple or pink subunit shown was Vps8 rather than Vps3.

Reviewer #2 (Remarks to the Author):

This manuscript reports a landmark cryoEM structure of the CORVET complex. CORVET, together with HOPS, are the main multisubunit tethering complexes that control trafficking and membrane fusion in the endolysosomal system. Technically impeccable and sure to be of great interest to the membrane trafficking field, this work should be published promptly.

I have a few suggestions.

Lines 100-102: The authors write: "As CORVET differs only at those positions from HOPS that control membrane specificity, we predict that the form and shape of the human CORVET and HOPS complexes will be similar if not identical to the yeast complexes." Aside from quibbles as to the difference between "form" and "shape", I don't understand how the second clause of this sentence follows from the first – could the authors please clarify?

Lines 183-200: I had difficulty following this paragraph – could the authors please consider rewriting for greater clarity?

Line 224: My best interpretation of Extended Data Fig. 7C is that Vps11 is missing (?!). But it would seem impossible that 11 is missing if the rest of the complex is intact. Is there a reason that authors don't use mass spec assay as in Ext Data Fig. 6? Perhaps this can be resolved by labeling the bands in Extended Data Fig. 7C.

Lines 228-232: Is it fair to surmise that they authors did not set up an in vitro fusion assay? Can they comment on why they didn't include a fusion assay in this work?

Lines 270-287: I'm having trouble reconciling the in vitro and in vivo consequences of the Vps11 Δ N and Vps18 Δ N mutations. Specifically, the deletions appear to have contradictory effects on tethering in vitro and endolysosomal trafficking in vivo: Vps11 Δ N displays wild type-level CORVET-mediated tethering in vitro but endosomal transport defects in vivo, whereas Vps18 Δ N abolishes tethering in vitro without apparently affecting endosomal transport in vivo. How do the authors think about these results? Furthermore, how do they fit with the fact that, in vivo, Vps11 Δ N and Vps18 Δ N would each compromise HOPS as well (see Shvarev 2022)?

Lines 295-298: The authors argue that the predicted Rab-interaction modes are similar to those of the HOPS complex and involve conserved residues known to mediate Rab-effector binding. I found it hard to validate this statement using the panels in Extended Data Fig. 9; for example, panel G only shows two sequences, and no panel presents the conserved residues mapped onto the structure. I would advocate for a more sophisticated analysis, if possible.

Lines 317-320: It would be helpful to point out in the legend for Fig. 6 that the SNAREs are not drawn to scale.

Lines 339-340: The authors rightly point out that the interaction between Vps8 and the Vps11 β -propeller is one of the, if not the, most striking differences between CORVET and HOPS. I am wondering whether they attempted to engineer mutations into this interface, and if so what did they learn? This would be a worthy, if not absolutely essential, addition to the manuscript.

We thank the editor for evaluating our manuscript and the reviewers for their positive and constructive feedback on our paper, as well as their insightful comments. We have addressed all of the concerns raised by the reviewers and have made the necessary modifications to the text and figures. Furthermore, we have made slight adjustments to the pdb file of the CORVET structure in response to a request from the wwPDB Biocuration Team, and have updated Table S3 and the figures in the manuscript accordingly.

REVIEWER COMMENTS

Reviewer #1 (Remarks to the Author):

The authors present the first cryoEM structure of the endosomal tethering complex CORVET. The resolution of the reconstruction is good enough that most of the complex can be built, excepting a highly flexible portion of Vps3. The authors are then able to compare and contrast the structure of CORVET to that of the related lysosomal tethering complex HOPS. They are able to explain how these structural differences correspond to the known functional differences between the complexes.

The cryoEM model is consistent with previously published data on Vps11 and Vps18 ts-alleles, and the authors make new truncation mutations based on the structure in order to probe the consequences of these mutations. They find that a mutant lacking the Vps11 beta-propeller increases the flexibility of Vps8 by determining an additional cryoEM structure.

The authors establish an in vitro vesicle tethering assay for CORVET, similar to the one established for the related complex HOPS. They use the assay to determine that CORVET-dependent liposome tethering requires Rab5/Vps21, PI3P, and lipid-packing defects.

They then determined that the Vps18 beta-propeller was required for CORVET function in the tethering assay, but the Vps11 beta-propeller was not. In contrast, they find that the Vps11 beta-propeller is required for CORVET function in cells but the Vps18 beta-propeller was not.

Finally, the authors present AlphaFold predictions of Rab5/Vps21 interactions with CORVET along with lower-resolution (negative stain) structural information of CORVET bound to liposomes via interactions with Rab5/Vps21, providing evidence for an “upright” positioning of CORVET on endosomal membranes.

Overall I found this to be an interesting and well-performed study. Given the fundamental importance of CORVET in cell biology, I think this should appeal to a broad readership. I have only minor suggestions for improvement.

1. The authors do not directly address the discrepancies between the results of the in vitro tethering assay and the in vivo functional tests regarding the Vps11 and Vps18 beta-propellers. One interpretation is that the in vitro tethering assay does not faithfully represent the situation in vivo. And while no in vitro assay can be expected to fully recapitulate all aspects of in vivo function, the authors are encouraged to mention in the discussion possible reasons for this discrepancy and potential avenues for improving the assay in the future.

We agree with the reviewer’s interpretation of these discrepancies, which suggests that the in vivo situation is more complex than the conditions in our in vitro tethering assay. For example, in vivo, Vps11 Δ N may impact not only CORVET but also HOPS, thereby causing the observed phenotype, while Vps18 Δ N may be complemented by additional cellular factors. We have also mentioned this point in the discussion as suggested by the reviewer (lines 360-369).

2. Lines 30 and 239 – “membrane packaging defects” should be “membrane packing defects” or maybe “lipid packing defects”

The text has been changed accordingly.

3. In Extended Figure S6C, E, and G, labels should be added for each subunit shown. I was initially confused because I thought the purple or pink subunit shown was Vps8 rather than Vps3.

The figure has been updated and the labels have been added.

Reviewer #2 (Remarks to the Author):

This manuscript reports a landmark cryoEM structure of the CORVET complex. CORVET, together with HOPS, are the main multisubunit tethering complexes that control trafficking and membrane fusion in the endolysosomal system. Technically impeccable and sure to be of great interest to the membrane trafficking field, this work should be published promptly.

I have a few suggestions.

Lines 100-102: The authors write: "As CORVET differs only at those positions from HOPS that control membrane specificity, we predict that the form and shape of the human CORVET and HOPS complexes will be similar if not identical to the yeast complexes." Aside from quibbles as to the difference between "form" and "shape", I don't understand how the second clause of this sentence follows from the first – could the authors please clarify?

We have rephrased the sentence accordingly for better clarity.

Lines 183-200: I had difficulty following this paragraph – could the authors please consider rewriting for greater clarity?

The paragraph has been revised as suggested by the reviewer.

Line 224: My best interpretation of Extended Data Fig. 7C is that Vps11 is missing (?). But it would seem impossible that 11 is missing if the rest of the complex is intact. Is there a reason that authors don't use mass spec assay as in Ext Data Fig. 6? Perhaps this can be resolved by labeling the bands in Extended Data Fig. 7C.

We are confident that mutated Vps11 is still present in the complex, as it is clearly recognisable in the cryo-EM structure (Figure 4E) obtained from the same sample used for SDS-PAGE (Extended Data Fig. 7C). Due to the mass reduction of mutant Vps11 compared to wild-type Vps11, it is aligned with Vps33 on SDS-PAGE. We have labelled the bands in Extended Data Fig. 7C as suggested.

Lines 228-232: Is it fair to surmise that they authors did not set up an in vitro fusion assay? Can they comment on why they didn't include a fusion assay in this work?

The primary focus of this work was on elucidating the structural aspects of CORVET and the functionality of the sample was verified through an in vitro tethering assay. We recognize the importance of an in vitro fusion assay for functional characterization of CORVET, however, our analysis revealed that CORVET requires specific parameters for correct functioning, such as a definite lipid composition, while it is likely that it recognizes the same SNAREs like HOPS as Vps33 is a shared subunit. Therefore, setting up of such an assay requires extensive additional investigation. We think that this should be the objective of a follow-up study.

Lines 270-287: I'm having trouble reconciling the in vitro and in vivo consequences of the Vps11 Δ N and Vps18 Δ N mutations. Specifically, the deletions appear to have contradictory effects on tethering in vitro and endolysosomal trafficking in vivo: Vps11 Δ N displays wild type-level CORVET-mediated tethering in vitro but endosomal transport defects in vivo, whereas Vps18 Δ N abolishes tethering in vitro without apparently affecting endosomal transport in vivo. How do the authors think about these results? Furthermore, how do they fit with the fact that, in vivo, Vps11 Δ N and Vps18 Δ N would each compromise HOPS as well (see Shvarev 2022)?

We have the following explanation for these results.

Both mutations, Vps11 Δ N and Vps18 Δ N, compromise HOPS in vitro, as we tested their functioning only by tethering and fusion assays (Shvarev et al. 2022). However, only Vps18 Δ N abolishes CORVET-mediated tethering in vitro. Our interpretation is that in both complexes, the Vps18 beta-propeller plays a crucial role in maintaining the integrity of the complex and in recognising and binding to the membrane, Rab GTPase, and SNAREs. In contrast, the Vps11 beta-propeller seems to be only essential in HOPS-mediated tethering and fusion. In CORVET, its role is likely compensated by other subunits, such as Vps8.

In the in vivo situation, we tested the importance of Vps11 Δ N and Vps18 Δ N mutants in endosomal transport. We assume that the observed phenotype results not only from the dysfunction of CORVET but also HOPS, thus compromising endosomal transport. On the other hand, it is possible that the Vps18 Δ N mutation is complemented by other components within the cell, which were not included in our in vitro assays.

We have also addressed this issue by expanding the discussion section (lines 360-369).

Lines 295-298: The authors argue that the predicted Rab-interaction modes are similar to those of the HOPS complex and involve conserved residues known to mediate Rab-effector binding. I found it hard to validate this statement using the panels in Extended Data Fig. 9; for example, panel G only shows two sequences, and no panel presents the conserved residues mapped onto the structure. I would advocate for a more sophisticated analysis, if possible.

The text and Extended Data Fig. 9 have been revised accordingly. We have included additional Rab proteins in the sequence alignment (panel G). Also, we have labelled the conserved residues of the predicted interaction interface for Ypt7 and Vps21 on the structural models (panels B, C, E, F) and sequences (panel G).

Lines 317-320: It would be helpful to point out in the legend for Fig. 6 that the SNAREs are not drawn to scale.

The figure legend has been changed accordingly (Line 321).

Lines 339-340: The authors rightly point out that the interaction between Vps8 and the Vps11 β -propeller is one of the, if not the, most striking differences between CORVET and HOPS. I am wondering whether they attempted

to engineer mutations into this interface, and if so what did they learn? This would be a worthy, if not absolutely essential, addition to the manuscript.

In this study, we analysed the functional role of Vps11-Vps8 interactions by focusing on the complete deletion of Vps11 β -propeller. The data obtained showed that this interface doesn't appear to affect CORVET-mediated tethering in vitro, but may be crucial for the functioning of CORVET in vivo (e.g. in fusion). However, we acknowledge that conducting additional experiments, such as improving the resolution of the structure in the region of the Vps11-Vps8 interface and engineering and analyzing corresponding point mutations, would further enhance our understanding of CORVET function. These experiments will be addressed in our future studies.

Reviewer #1 (Remarks to the Author):

The authors have addressed my minor concerns and I am strongly supportive of publication of this outstanding manuscript.

Reviewer #2 (Remarks to the Author):

The authors have addressed my concerns satisfactorily; I think this important manuscript is ready for publication!